# Ln^3+^-Induced Diblock Copolymeric Aggregates for Fully Flexible Tunable White-Light Materials

**DOI:** 10.3390/nano9030363

**Published:** 2019-03-05

**Authors:** Xinzhi Wang, Jianguo Tang, Guanghui Wang, Wei Wang, Junjie Ren, Wei Ding, Xinbo Zhang, Yao Wang, Wenfei Shen, Linjun Huang, Laurence A. Belfiore

**Affiliations:** 1Institute of Hybrid Materials, National Center of International Joint Research for Hybrid Materials Technology, National Base of International Sci. & Tech. Cooperation on Hybrid Materials, Qingdao University, 308 Ningxia Road, Qingdao 266071, China; wangxinzhi1988123@163.com (X.W.); guanghuiwang@qdu.edu.cn (G.W.); wangwei040901@163.com (W.W.); renjunjie0718@163.com (J.R.); dingweiedu@163.com (W.D.); znpt95@gmail.com (X.Z.); wangyaoqdu@126.com (Y.W.); jiushiwo5698@126.com (W.S.); newboy66@126.com (L.H.); 2Department of Chemical and Biological Engineering, Colorado State University, Fort Collins, CO 80523, USA

**Keywords:** nano-aggregates, diblock copolymer, fully flexible, white-light

## Abstract

In this research contribution, nano-aggregates have been fabricated by introducing lanthanide (Ln^3+^) ions into solutions of amphiphilic diblock copolymers of polystyrene-b-poly (acrylic acid) (PS-b-PAA). The coordination of acrylic acid segments to lanthanide cations induces diblock copolymer (BCPs) self-assembly in order to design stable white luminescent hybrid nanoparticles with fine uniform particle size. The introduction of Ln^3+^ ions (Eu^3+^ and Tb^3+^) bestows the micelles, precisely white light, upon excitation of 342 nm. Lanthanide coordination cross-linking of poly (acrylic acid) segments, or blocks, endows the micelles higher thermal stability than that of BCPs micelles without cross-linking. As the most important key point of this work, the regular and stable nano-particles with high emission quality can make fully flexible electroluminescent devices with self-formation or uncoordinated into polymer hosts. Instead of inorganic luminescent nanoparticles with hard cores, this method can potentially apply for fully flexible white-light emitting diodes (FFWLEDs).

## 1. Introduction

Polymer-based light emission materials has gained hugely increased attention in recent years because of their flexible property to meet roll-up/spread applications, such as full color displays and photoelectronic devices [1,2,3,4]. In particular, white-light-emitting diodes (WLEDs) are of higher concern due to their undeniable energy efficiency to replace incandescent bulbs [5,6]. Presently, white light generation depends on the color mixing principles of trichromatic (red, green, and blue) or dichromatic (yellow, blue) lights [7,8,9], thus there are two characteristic modes to fabricate white-light-emitting materials in the literature. The first is blue light as a light resource to excite yellow phosphors, which is widely used due to the technique’s simplicity and low cost. The commendably realized white LED, YAG: Ce, is used as yellow phosphor. Unfortunately, it has many limitations, such as low luminous efficiency and color-rendering index [10]. The second is UV-light excited phosphors to emit red, green, and blue light, which can solve the above limitations and can improve the stability of light color. However, there is extremely high requirement for the purity of red, green, and blue emissions. Therefore, the scientific target is to find stable, narrow-emitting and highly luminous, efficient materials [11,12]. 

Lanthanide complexes have unique luminescence properties that are represented by narrow line-like emission bands in the visible range (400–800 nm) with high color purity due to the unique 4f electronic states of the lanthanide ions. These 4f electronic states of lanthanide ions are shielded by their outer 5s and 5p electronic orbitals, and the partially electronic transitions between 4f orbital electronic states are strictly forbidden [13,14]. Eu^3+^ and Tb^3+^ ions are well known for their sharp emission bands locating around 612 nm and 545 nm, respectively [15,16]. With the interference of suitable ligands, Eu^3+^ and Tb^3+^ complexes usually exhibit prominent red and green light. At the same time, the addition of blue emission can be obtained from organic ligands or polymers [17,18]. Therefore, the chemical structure and energy levels of organic ligands should be considered carefully, which is a great challenge to obtain highly bright and pure white light. Organic ligands with the function as antenna can obviously improve the absorbance of ultraviolet radiation of rare-earth ions [19]. The frequently used ligands are exemplified by conjugated small molecules like β-diketone and its derivatives.

Polymeric hosts can donate the flexible nature into lanthanide complexes to get processability [20,21], which is a necessary property to obtain films, devices, and so on. The simple and traditional method is to dope luminescent species into polymer matrices, making them blend together. Its limitation is identified as having the worst compatibility between the Ln^3+^ luminescence center and polymeric matrices, which leads to the phase separation phenomenon. The second method is to make luminescence ion coordination with specific groups along the polymer chains to form macromolecule-Ln^3+^ blends that can improve the compatibility [22,23], but the macromolecule-Ln^3+^ blends normally form solid precipitates from the solution, which leads to processing difficulty for further uses.

Therefore, it is important to find a Ln^3+^-containing polymer hybrid system with both good compatibility and processable performances for flexible luminescence materials. Amphiphilic diblock copolymers can form different aggregates by self-assembly in selective solvents [24]. One segment of amphiphilic diblock copolymers is soluble and the other is not. The insoluble segments act as the core surrounded by the soluble segments, which are highly swollen by the solvent [25]. Amphiphilic diblock copolymers can be synthesized by the reversible addition-fragmentation chain transfer (RAFT), a living/controlled free radical polymerization providing very low polydispersity of polymerization degree [26,27]. 

Polymeric aggregates (BCPs-Ln^3+^-Phen) containing Eu^3+^ and Tb^3+^ ions are successfully synthesized as illustrated in Figure 1. The BCPs are synthesized by RAFT polymerization. The rare earth ions are introduced into BCPs solution and coordinate with acrylic acid groups from different polymeric chains, leading to the formation of cross-linked PAA-Ln^3+^ domains. This reaction does not affect another segment, PS, which maintains its compatibility with the organic solvent. Consequently, BCP-Ln^3+^ micelles are formed with a PAA-Ln^3+^ core and PS shell. Therefore, the stable micelles including Eu^3+^ and Tb^3+^ complexes with organic small conjugate ligands can provide tunable photoluminescence property to cover blue, green, and red under a single 342 nm excitation. Tricolor lights can coordinate white light. So, BCPs-Ln^3+^-Phen emits bright white light. These lanthanide complexes are dissolved in solution to evaluate the optical properties and surface morphologies. The blue, green, and red light colors of BCPs-Ln^3+^-Phen are generated by tuning the ratios of Eu^3+^ and Tb^3+^ ions and excitation wavelengths, and CIE coordinates are located in the white light region. On the other hand, the cross-linking endows the micelles higher structural stability than that of BCPs micelles without cross-linking. Therefore, the white light solid micelles BCPs-Ln^3+^-Phen are remarkable luminescent materials for designing and fabricating WLEDs. Ln^3+^-Phen can be obtained by tuning the ratios of Eu^3+^ and Tb^3+^ ions and excitation wavelength.

## 2. Materials and Methods 

### 2.1. Materials

Styrene, acrylic acid, and 1,10-phenanthroline (phen) were purchased from Aladdin (Shanghai, China). Polymerization inhibitor in styrene and acrylic acid were removed before use. 2,2-azobisisobutyronitrile (AIBN) was obtained from Alfa Aesar, and it needed to recrystallize from ethanol. The solvents of dichloromethane, methanol, petroleum ether, and N, N-dimethyl formate (DMF) are analytical solvents that can be used directly. Eu_2_O_3_ and Tb_4_O_7_ (≥99.9%) were purchased from Shandong Desheng new materials Company Limited. After that, DMF was used as solvent to prepare the Eu^3+^ and Tb^3+^ solution, respectively. 

### 2.2. Characterization

The measurement and characteristics of equipment in the article are displayed in Table 1. 

### 2.3. Preparation of Diblock Copolymers: Polystyrene-Block-Polyacrylic Acid (PS-b-PAA)

#### 2.3.1. Synthesis of RAFT Agent

The synthesis of the RAFT agent is referred to elsewhere in the literature [28]. ^1^HNMR spectra of (a) 4-cyano-4-ethyl trithiopentanoic acid and (b) RAFT agent are shown in Appendix A.

#### 2.3.2. Preparation of Diblock Copolymers PS-b-PAA

The synthesis of BCPs by RAFT polymerization was based on a previously reported method with sight modifications [29]. Amounts of 1.697 g styrene, 0.192 g RAFT, 0.029 g AIBN, and 5 mL 1,4-dioxane were introduced into a round-bottom flask, and the mixture solution was stirred for 30 min under nitrogen. Then, the mixture solution was heated at 80 °C for 10 h. The 1H NMR spectrum of PS reaction mixture after polymerization is shown in Figure 2a. The PS solid was obtained by using methanol to wash three times. PS was dried under vacuum for 24 h. Amounts of 1.012 g PS, 0.023 g AIBN, 0.641 g acrylic acid (AA), and 10 mL 1,4-dioxane were introduced into a round-bottom flask, then the mixture solution was stirred for 30 min under nitrogen. Finally, the mixture solution was heated at 65 °C for 10 h to prepare the PS-b-PAA solution. The PS-b-PAA solid was obtained to use petroleum ether to wash three times. PS-b-PAA was dried under vacuum at ambient temperature for 24 h. The ^1^HNMR spectrum of PS_21_-b-PAA_14_ is illustrated in Figure 2b.

#### 2.3.3. Synthesis of Eu^3+^/Tb^3+^ Complexes

##### (1) Synthesis of EuCl_3_·6H_2_O and TbCl_3_·6H_2_O

An amount of 1.00 g Eu_2_O_3_ was dissolved in 2.85 mL HCl (6 mol/L) and heated at 50 °C for 1 day, and then it was gradually cooled to room temperature, resulting in colorless crystals that were purified by using THF to wash and were dried at 40 °C in a vacuum drying oven. The synthesis of TbCl_3_·6H_2_O was the same with EuCl_3_·6H_2_O. 

##### (2) Synthesis of Eu^3+^/Tb^3+^ Complexes

PS-b-PAA, EuCl_3_·6H_2_O, and TbCl_3_·6H_2_O were dissolved in DMF at the AA, Eu^3+^, and Tb^3+^ concentration of 2 × 10^−2^ mol/L respectively. The antennae “ligand” 1,10-phenanthroline (phen) and 5 mL PS-b-PAA solution were added in a three-necked flask equipped with reflux. EuCl_3_ and TbCl_3_ solution were added dropwise into this solution at 60 °C. The reaction mixture was diluted in DMF. Five samples with different Eu^3+^/Tb^3+^ molar ratios were prepared, as shown in Appendix A. Then, the reaction was allowed to develop for 8 h until the uniform fluorescent aggregates were obtained.

## 3. Results and Discussion

### 3.1. The Structure of Polymeric Aggregates 

The absorption band for BCPs ranges from 260 nm to 340 nm and the maximum absorption is at 268 nm, which is primarily attributed to the effect of the π-π* transition of benzene ring of PS-b-PAA [30]. Comparing the absorption band of Ln^3+^-BCPs with BCPs, the maximum absorption at 268 nm becomes stronger. On the other hand, for free Phen, there is strong absorption band at 260–340 nm, including two absorption peaks at 294 nm and 324 nm, respectively. They can be ascribed to π-π* transition and n-π* transition respectively [31]. The peak profiles of Ln^3+^-Phen and BCPs-Ln^3+^-Phen are almost similar to ligand Phen, excepting the red shift of their absorption peaks. The strong absorption peaks of Ln^3+^-Phen and BCPs-Ln^3+^-Phen are at 297 nm and 299 nm, respectively. Both of them have red shift compared with Phen, which is attributed to the coordination of Ln^3+^ ions with N atoms of Phen. In addition, the strong absorption of BCPs-Ln^3+^-Phen, compared with extremely weak absorption of BCP-Ln^3+^, indicates that the UV-vis absorption mainly comes from the Phen ligand.

The infrared spectra of BCPs and BCPs-Ln^3+^-Phen are shown in Figure 3b. Two absorption bands appear at 2920 cm^−1^ and 2890 cm^−1^. The former is attributed to the stretching vibrations of C–H bonds in the groups –CH_2_, and the latter is assigned to the stretching vibrations of C–H bonds in the groups –CH. The absorption at 1700 cm^−1^ reflects the stretching vibration of C=O bonds [32]. The peaks at 1406 and 1296 cm^−1^ may be caused by vibration coupling of O-H and C-O in carboxyl groups. Additionally, the absorption bands of the monosubstituted benzene ring appear at 760 and 697 cm^−1^. These spectra confirm that BCPs were synthesized successfully. In the spectrum of BCPs-Ln^3+^-Phen, the absorption band of C=O bonds has disappeared, while there are two strong absorption peaks in 1590 and 1410 cm^−1^ compared with the absorption peaks in the spectrum of BCPs. This is because the H+ in the carboxylic acids are replaced by Ln^3+^ ions to form carboxylate, and the C=O and C-O connected to the same carbon atom are homogenized to be two C

O [33,34]. These two bonds connected on the same carbon atom have similar vibration frequencies, which can form two absorption peaks at 1590 and 1410 cm^−1^ ascribed to the occurrence of vibration coupling and asymmetric vibration and symmetric vibration of C

O, respectively. In general, the spectrum indicates that the O atom of the carboxyl groups has coordinated with Ln^3+^ ions.

#### 3.1.1. X-ray Photoelectron Spectroscopy (XPS)

X-ray photoelectron spectroscopy (XPS) was carried out to further examine the complementary information of BCPs-Ln^3+^-Phen. The XPS spectra of BCPs-Ln^3+^-Phen with the reference binding energy at 284.6 eV for the C 1s region is shown in Figure 4a. The peaks show the electronic states of Tb 3d, Eu 3d, O 1s, N 1s, C 1s, respectively.

Figure 4b shows the binding energy peaks of N 1s. Compared with Phen, the electronic binding energy of N 1s in BCPs-Ln^3+^-Phen exhibits apparent variations. The N 1s peak shifts toward higher binding energy about 0.7 eV, whereas the peak becomes weaker, implying that the vicinity environment around N atom has changed. Because the electron pair of N atom coordinates with Ln^3+^ ions, the N atom donates electron to the outer empty orbital of Ln^3+^ ion, leading to the increase of binding energy [35]. The O 1s spectra of BCPs and BCPs-Ln^3+^-Phen is displayed in Figure 4c,d, respectively. Their peaks can be divided into two overlapping peaks, which are attributed to the presence of two kinds of O atoms in carboxyl groups. The peaks of binding energy are 532.0 and 533.1 eV for BCPs and 532.2 and 533.4 eV for BCPs-Ln^3+^-Phen, which are ascribed to O atoms of C=O and C-O, respectively. The binding energy has a certain increase after coordination. These changes provide further evidence for N and O atoms coordination with Ln^3+^ ions.

### 3.2. Morphologies of Polymeric Assemblies.

To gain morphologies of polymeric aggregates, we characterized the BCPs-Ln^3+^-Phen reaction solution by transmission electron microscopy (TEM). TEM images display a relatively weak contrast, excepting the size of complexes particles. BCPs-Ln^3+^-Phen complexes are uniform and regular spheres. The reason may be that the coordination bonds between Eu^3+^, Tb^3+^ ions and carboxyl groups along the PAA chains have formed, and every Eu^3+^ and Tb^3+^ ions can form coordination bonds with two or more carboxyl groups belonging to different PAA chains in this diblock copolymer. As a result, carboxyl groups at the same polymer chain may react with different Eu^3+^ and Tb^3+^ ions, and every Eu^3+^ and Tb^3+^ ions may be coordinated with different polymer chains cross-linking PAA network, which is less soluble to DMF than the other segment, PS, which can keep good solubility to DMF solvent. Eu^3+^ and Tb^3+^ ions induce PS-b-PAA to form self-assembled polymeric aggregates that may minimize their surface energies as much as possible, just aiming to exist steadily and disperse homogeneously in DMF solution. Herein, BCPs-Ln^3+^-Phen form sphere aggregates.

Figure 5 shows different molar ratios of Eu^3+^ and Tb^3+^ ions in BCPs-Ln^3+^-Phen complexes. Eu^3+^/Tb^3+^ molar ratios of the samples (a), (b), (c), (d), and (e) are 3:1, 2:1, 1:1, 1:2, and 1:3, respectively. The average size of these micelles structures are: (a) 40–80 nm, (b) 130–170 nm, (c) 280–350 nm, (d) 100–200 nm, and (e) 50–100 nm. The sizes of micelles in samples (a) and (e) are much smaller, but that in sample (c) is much bigger. The influence of sample composition on micelle sizes can be ascribed to the different contents of Ln^3+^ ions. With the increase of Ln^3+^ ions, more Ln^3+^ ions take part in the coordination and the cross-linking densities among the different PAA segments increases. Therefore, the micelles become tighter, which makes the size smaller.

Elemental analyses of the BCP-Ln^3+^-Phen with different molar ratios of Eu^3+^ and Tb^3+^ ions are also used to affirm the results. From Table 2, we can see that the percentages of Eu^3+^ and Tb^3+^ atoms are essentially in agreement with the molar ratios of additions. The results confirm that the sizes of micelles are affected by the incorporated Ln^3+^ ions.

The morphological structure of the as-prepared BCPs-Ln^3+^-Phen are investigated by STEM and the energy dispersive X-ray (EDX) mappings are shown in Figure 6. From the EDX mapping image, we can observe the elements of N (yellow), O (blue), Eu^3+^ (red), and Tb^3+^ (green) are mostly distributed in the interior of the particle, proving that Ln^3+^ ions induce the BCP self-assembly to form the spherical nanoparticles.

EDS spectrum of BCPs-Ln^3+^-Phen (Eu:Tb = 1:1) is shown in Appendix A. EDS can detect the existence of various elements accurately. N, O, Eu, and Tb elements can be found in the spectrum, which further confirms the existence of the elements in the nanoparticles.

### 3.3. Fluorescence Property and White Light Performance

The photoluminescent property of samples containing different molar ratio of Ln^3+^ ions are shown in Figure 7. Excitation spectra is shown in (a) and the range of excitation wavelength is from 290 nm to 360 nm. As is well known, Ln^3+^ ions have extremely low molar absorption from 200 nm to 400 nm, which is ascribed to the forbiddance of electronic transitions between the 4f electronic states. Phen can be used as the “antenna” ligand due to its excellent molar absorptivity in the range. Because there is a suitable energy gap between trivalent state level of Phen and the Ln^3+^ ions emissive states, Ln^3+^ ions can effectively obtain energy from ligand under the action of appropriate antenna ligand. 

The emission spectra display six peaks at 425 nm (the blue emission), 491 nm, 545 nm (the green emission), 592 nm, 615 nm (the red emission), and 697 nm, respectively. The stable emission wavelengths of Eu^3+^ ions and Tb^3+^ ions ascribe to that the 4f electrons of Eu^3+^ ions and Tb^3+^ ions are shielded by 5s5p electrons that makes characteristic emission of Eu^3+^ ions and Tb^3+^ ions be insensitive to the environment. The blue emission mechanism is fully different with Eu^3+^ ions and Tb^3+^ ions, which is attributable to the π-π* transition of Phen. In addition, their emission intensity has fluent change with the mixing of molar ratios of Eu^3+^ ions and Tb^3+^ ions from 3:1 to 1:3.

Hence, a variety of colors are obtained by adjusting ratios of Eu^3+^ ions and Tb^3+^ ions as calculated by CIE coordinates (Figure 7c). The calculated CIE coordinates of complexes a, b, c, d, e are at (0.355, 0.290), (0.347, 0.303), (0.342, 0.346), (0.300, 0.363), and (0.326, 0.432), respectively excited at 340 nm. It is noteworthy that the CIE coordinate of complex c excited at 340 nm is close to the ideal coordinate for pure white light [36,37].

TG analysis (Figure 7d) shows that pure BCPs exhibit three major decomposition events. The first step is attributed to little solvent in BCPs, the second step is due to the loss of side chains along the polymer backbone, and the third step is associated with the decomposition of the BCPs backbone [38]. 

Meanwhile, for the complex BCPs-Ln^3+^-Phen, the weight loss begins from 240 °C in Table 3. Compared with the TG curves of pure BCPs, complex BCPs-Ln^3+^-Phen has an improved thermal stability with a 48 °C increment, showing that BCPs-Ln^3+^-Phen has significantly good thermal stability.

In order to obtain more pure white light, we changed the excitation wavelength between 330 nm and 350 nm to investigate the emission behavior of complex c because the optimum excitation wavelength is 340 nm in Figure 8a, and the intensity of the characteristic fluorescence shows different change as shown in Figure 8b. The fluorescence intensity became stronger from 400 nm to 430 nm (blue emission) when the excitation wavelength changed from 330 nm to 350 nm, while the characteristic peaks at 545 nm (green emission) and 615 nm (red emission) became stronger along with the change of excitation wavelength from 330 nm to 345 nm, then the characteristic peaks also became lower. The corresponding CIE chromaticity diagram is shown in Figure 8c. The value of the corresponding CIE chromaticity coordinates of complex c decreases and the coordinates (Table 4) vary from yellow emission (0.404, 0.416) to blue light emission (0.258, 0.239) including the white-light region. Meanwhile, according to the 1931 CIE coordinate diagram [39,40], the CIE coordinates (0.335, 0.332) of BCPs-Ln^3+^-Phen complexes with a 1:1 molar ratio of Eu^3+^ to Tb^3+^ is very close to the ideal white-light (x = 0.333, y = 0.333) upon excitation of 342 nm.

In order to further investigate another important luminescence property of BCPs-Ln^3+^-Phen, the emission decay curves are presented in Figure 8d from the ^5^D_0_ and ^5^D_4_ states were measured by monitoring the ^5^D_0_-^7^F_2_ and ^5^D_4_-^7^F_5_ emission lines respectively. The luminescence life time of Eu^3+^ ions and Tb^3+^ ions are 485 μs and 521 μs.

It is well known that the organic ligands absorb UV light, transiting to the single excited state (^1^ππ*) and then to the triplet state (^3^ππ*), and the last energy transfer from the triplet state to the resonance level of the Ln(III) ion is an important factor influencing the luminescent performance of lanthanide complexes. Figure 9 is energy level scheme of complex c. The energy level of a single excited state is 30303 cm^−1^, which was obtained from the wavelength (330 nm) of UV-vis absorption in Figure 3a. The energy level of triplet is 23810 cm^−1^ (420 nm), ^5^D_0_ of Eu^3+^ is 17301 cm^−1^ (578 nm), and ^5^D_4_ of Tb^3+^ is 20492 cm^−1^ (488 nm). When ΔE (^1^ππ* − ^3^ππ*) is at least 5000 cm^−1^, ligand to metal energy transfer becomes effective [31]. Therefore, energy transfer happens from ligands to Eu^3+^ and Tb^3+^ ions and energy transfer from ligands to Eu^3+^ ions is more efficient than to Tb^3+^ ions.

## 4. Conclusions

In conclusion, we have synthesized the novel and flexible fluorescent polymeric micelles (BCPs-Ln^3+^-Phen) from Eu^3+^ and Tb^3+^ ions coordinated amphiphilic diblock copolymers (PS-b-PAA) which can be used as excellent flexible materials. Relative to pure diblock copolymers, BCPs-Ln^3+^-Phen complexes display more stable thermal stability with a decomposition temperature of 240 °C, which fulfills the requirement of the LED emitting layer.

When complexes are excited at 342 nm, BCPs-Ln^3+^-Phen emits blue light at 425 nm, green light at 545 nm, and red light at 615 nm. Weaker emissions are observed at 488 nm, 592 nm, and 697 nm. According to the 1931 CIE coordinate diagram, the CIE coordinates (0.335, 0.332) of BCPs-Ln^3+^-Phen complexes with a 1:1 molar ratio of europium(III) to terbium(III), and excitation at 342 nm is extremely close to ideal white-light (x = 0.333, y = 0.333). The luminescence life times of Eu^3+^ ions and Tb^3+^ ions are 485 μs and 521 μs, respectively. Considering all of these optoelectronic properties, the advantages of BCPs-Ln^3+^-Phen complexes are promising candidates for practical application for fully flexible WLEDs (FFWLEDs). This study provides a direction for further research of rare earth complexes in fully flexible white-light emitting diodes (FFWLEDs).

## Figures and Tables

**Figure 1 nanomaterials-09-00363-f001:**
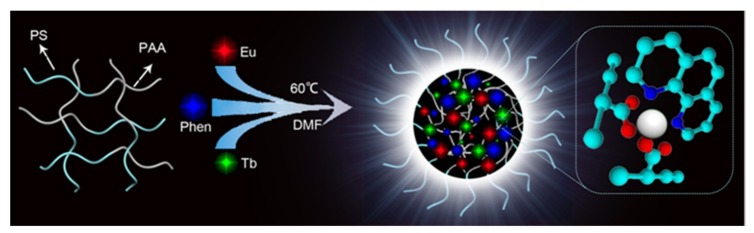
Schematic diagram of BCPs-Ln^3+^-Phen. BCPs, diblock copolymer.

**Figure 2 nanomaterials-09-00363-f002:**
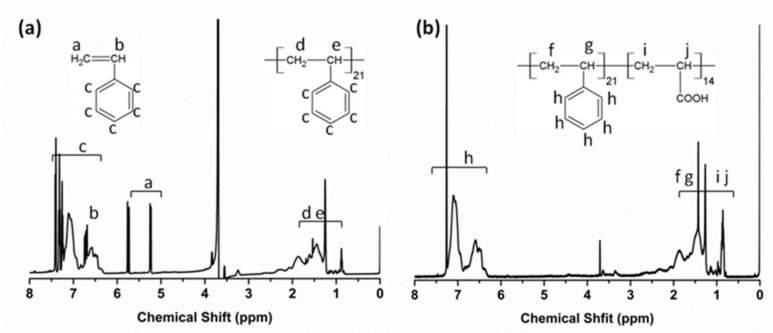
^1^H NMR spectra of PS reaction mixture after polymerization (**a**) and PS_21_-b-PAA_14_ (**b**).

**Figure 3 nanomaterials-09-00363-f003:**
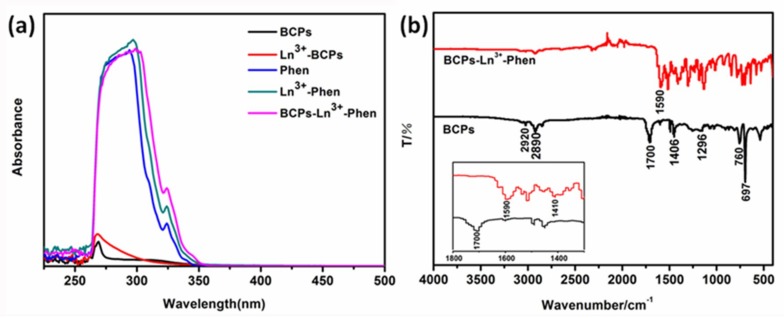
(**a**) UV-vis absorption spectra of BCPs, Ln^3+^-BCPs, Phen Ln^3+^-Phen, and BCPs -Ln^3+^-Phen; (**b**) FTIR spectra of BCPs and BCPs-Ln^3+^-Phen. Inset: the magnify spectra of (b).

**Figure 4 nanomaterials-09-00363-f004:**
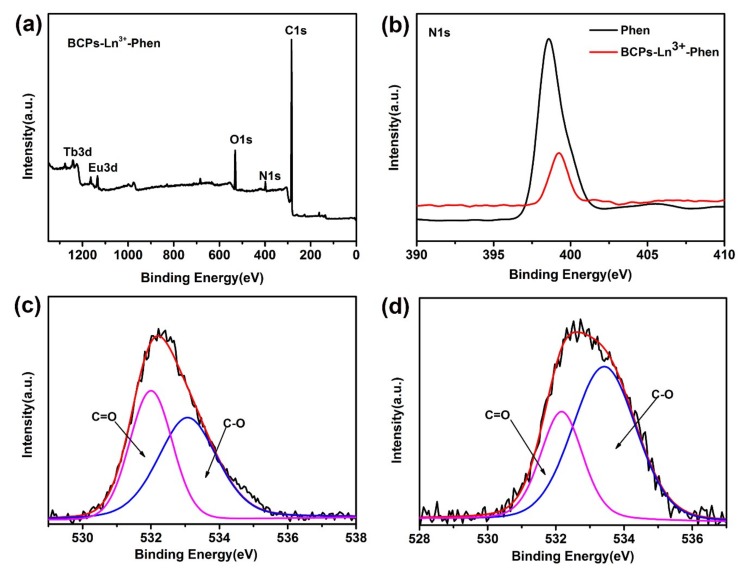
(**a**) XPS survey of BCP-Ln^3+^-Phen; (**b**) XPS spectra of the N1s region of Phen and BCPs-Ln^3+^-Phen; (**c**) XPS spectra of the O1s region of BCPs; (**d**) XPS spectra of the O1s region of BCPs-Ln^3+^-Phen.

**Figure 5 nanomaterials-09-00363-f005:**
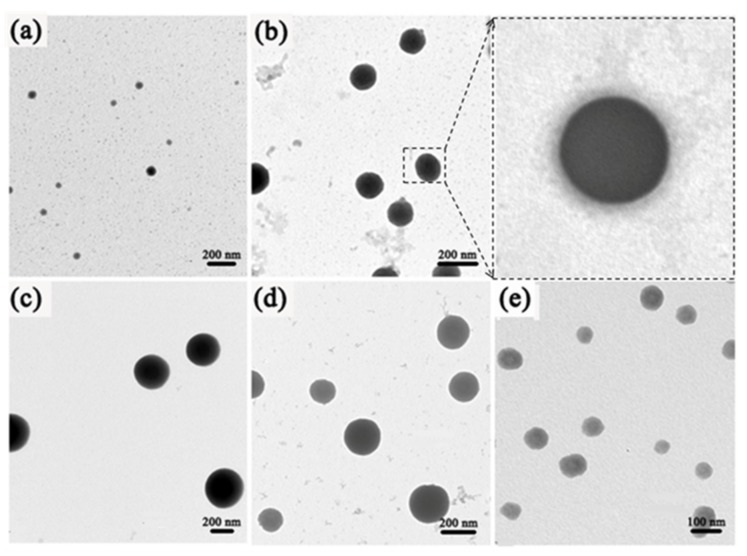
TEM images of samples with Eu^3+^/Tb^3+^ molar ratios of (**a**) Eu^3+^/Tb^3+^ = 3:1, (**b**) Eu^3+^/Tb^3+^ = 2:1 and its magnified image, (**c**) Eu^3+^/Tb^3+^ = 1:1, (**d**) Eu^3+^/Tb^3+^ = 1:2, (**e**) Eu^3+^/Tb^3+^ = 1:3.

**Figure 6 nanomaterials-09-00363-f006:**
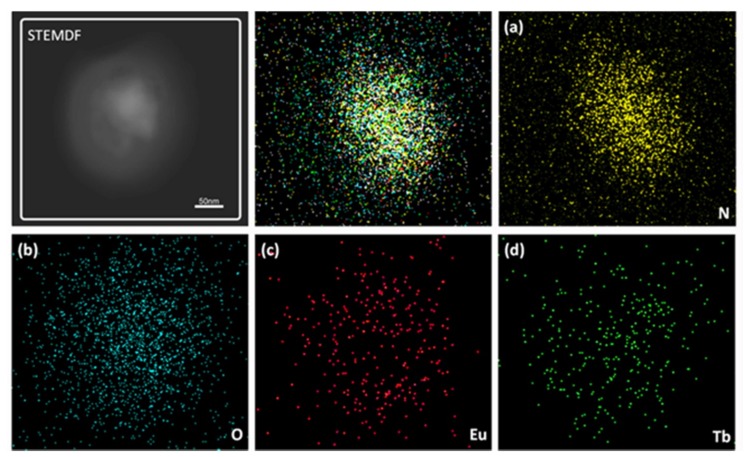
STEM dark-field (DF) image, (**a**–**d**) elemental mapping of N, O, Eu, Tb elements of BCPs-Ln^3+^-Phen, individually.

**Figure 7 nanomaterials-09-00363-f007:**
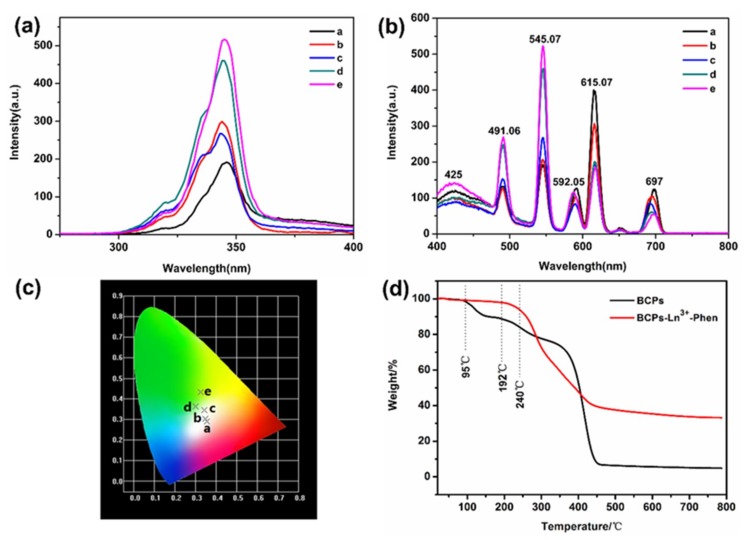
Fluorescence property, white light performance, and thermal stability of BCPs-Ln^3+^-Phen (**a**) Excitation spectra and (**b**) emission spectra of samples containing different molar ratio of Ln^3+^ ions, Eu^3+^/Tb^3+^ = 3:1 Eu^3+^/Tb^3+^ = 2:1, Eu^3+^/Tb^3+^ = 1:1, Eu^3+^/Tb^3+^ = 1:2, Eu^3+^/Tb^3+^ = 1:3, (λ_ex_ = 340 nm). (**c**) The CIE chromaticity coordinates of complexes a–e excited by 340 nm [a (0.355, 0.290), b (0.347, 0.303), c (0.342, 0.346), d (0.300, 0.363), e (0.326, 0.432)]. (**d**) TG curves of BCPs and BCPs-Ln^3+^-Phen.

**Figure 8 nanomaterials-09-00363-f008:**
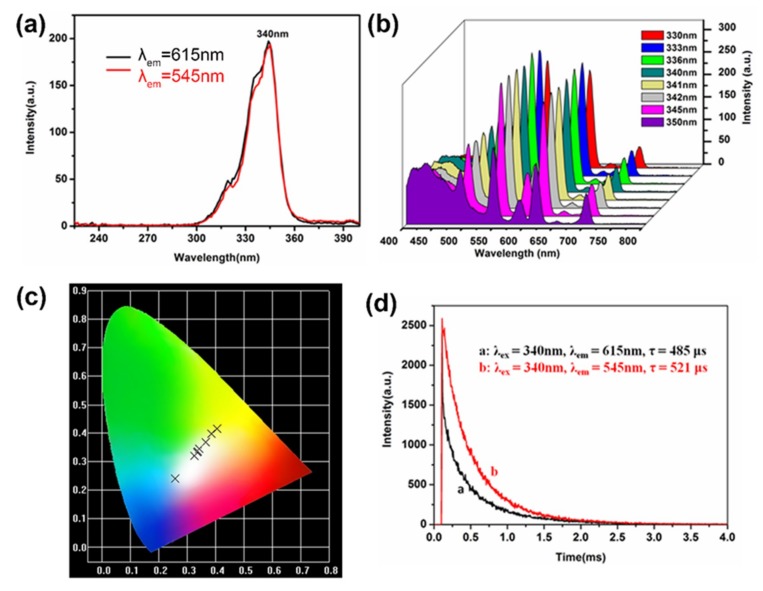
(a) Excitation spectra of complex c with two different emission wavelengths. Emission spectra (b) and CIE chromaticity diagram (c) of complex c by excitation wavelength from 330 to 350 nm (shown in fork symbols from right to left, referring to the excitation wavelength at 330, 333, 336, 340, 341, 342, 345, 350 nm, respectively). (d) The fluorescence decay curves of BCPs-Ln^3+^-Phen (a. ^5^D_0_-^7^F_2_ transition of Eu^3+^ ions, b. ^5^D_4_-^7^F_5_ transition of Tb^3+^ ions).

**Figure 9 nanomaterials-09-00363-f009:**
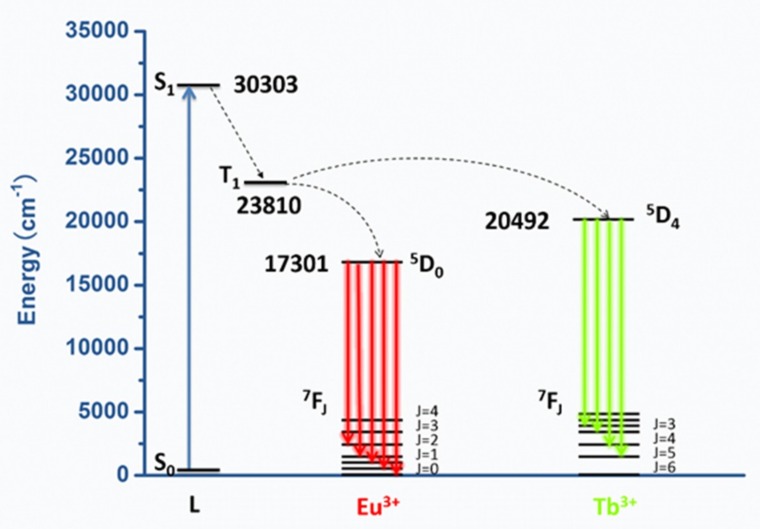
Energy level scheme showing energy transfer processes in complex c.

**Table 1 nanomaterials-09-00363-t001:** Instrumentation.

Measurement	Characteristics of Equipment
1H NMR spectra	JNM-ECP600 (600 MHz) spectrometer JEOL Ltd, Kyoto, Japan, CDCl_3_ as the solvent
UV/vis spectra	UV755B spectrophotometer, Youke, Shanghai, China at room temperature
NIR spectra	infrared spectrometer, Nicolet 5700, Thermo Fisher Scientific, Varian, Inc., Palo Alto, CA, USA
XPS spectra	Kratos spectrometer, XSAM-800, ESCALAB, London, UK
TEM micrographs	JEM-1200EX electron microscope, JEOL Ltd., Kyoto, Japan
EDS mapping	JEM-2100 high-resolution transmission electron microscope, JEOL, Kyoto, Japan
Fluorescent spectra	fluorescence spectrophotometer (Varian, Inc., Palo Alto, CA, USA) at ambient temperature
TG curves	Thermo gravimetric analysis, SII TG/DTA 6300, Waltham, MA, USA
Lifetime	Fluorescence Spectrometer, FLS980, Edinburgh, UK

**Table 2 nanomaterials-09-00363-t002:** Elemental analyses of the BCP-Ln^3+^-Phen complex a–e.

Samples	Elements			
	C (Atomic%)	O (Atomic%)	Eu (Atomic%)	Tb (Atomic%)
a	86.02	13.35	0.45	0.19
b	87.01	12.70	0.19	0.10
c	77.68	22.10	0.13	0.009
d	92.48	7.14	0.11	0.27
e	88.85	10.70	0.11	0.33

**Table 3 nanomaterials-09-00363-t003:** Decomposition temperature of BCPs and BCPs-Ln^3+^-Phen.

Samples	Decomposition Temperature (°C)
BCPs	192
BCPs-Ln^3+^-Phen	240

**Table 4 nanomaterials-09-00363-t004:** CIE coordinates of BCPs-Ln^3+^-Phen under different excitation wavelengths.

Excitation Wavelength (nm)	CIE Coordinates
330	(0.404, 0.416)
333	(0.386, 0.396)
336	(0.364, 0.368)
340	(0.341, 0.338)
341	(0.337, 0.334)
342	(0.335, 0.332)
345	(0.324, 0.320)
350	(0.258, 0.239)

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
