# Peer review of "Ln^3+^-Induced Diblock Copolymeric Aggregates for Fully Flexible Tunable White-Light Materials"

_nanomaterials, 2019, doi:10.3390/nano9030363_

Reviewer 1 Report

The main purpose of this manuscript it that lanthanide ion is embedded into amphiphilic deblock copolymers and its application to white light-emitting diode. The contents are almost acceptable however following content should be revised.

1.      p.2, line 75. Eu3+ -> Eu3+

2.      p.3, line 103. (JEOL, Japan).Samples -> (JEOL, Japan). Samples

3.      p.5, line 159. 2920 cm-1 and 2890 cm-1 -> 2920 cm-1 and 2890 cm-1

4.      p.6, line 181. The XPS spectrum in Fig. 4(b) indicates there are two peaks for BCPs-Ln3+-Phen. The peak fitting is needed to understand in detail.

5.      p.7, line 203. I agree that the size of micelles is affected by the incorporated ion. The composition measurement is needed to explain about this.

6.      I found several mistakes about the English grammar. Please check again.

Author Response

The main purpose of this manuscript it that lanthanide ion is embedded into amphiphilic deblock copolymers and its application to white light-emitting diode. The contents are almost acceptable however following content should be revised.

Point 1:      p.2, line 75. Eu3+ -> Eu3+

Point 2:      p.3, line 103. (JEOL, Japan).Samples -> (JEOL, Japan). Samples

Point 3:      p.5, line 159. 2920 cm-1 and 2890 cm-1 -> 2920 cm-1 and 2890 cm-1

Point 6:      I found several mistakes about the English grammar. Please check again.

Response: Thank you for your advices, I have revised the contents of 1, 2, 3 and 6 in the manuscript.

Point 4:      p.6, line 181. The XPS spectrum in Fig. 4(b) indicates there are two peaks for BCPs-Ln3+-Phen. The peak fitting is needed to understand in detail.

Response 4:

Fig. 4b shows the binding energy peaks of N 1s. Compared with Phen, the electronic binding energy of N 1s in BCPs-Ln3+-Phen exhibits apparent variations. The N 1s peak shifts toward higher binding energy about 0.7 eV, whereas the peak becomes weaker, implying that the vicinity environment around N atom has changed. Because the electron pair of N atom coordinates with Ln3+ ions, the N atom donates electron to the outer empty orbital of Ln3+ ion, leading to the increase of binding energy [35]. The O 1s spectra of BCPs and BCPs-Ln3+-Phen is displayed in Fig. 4c-d, respectively. Their peaks can be divided into two overlapping peaks, which are attributed to the presence of two kinds of O atoms in carboxyl groups. The peaks of binding energy are 532.0 and 533.1 eV for BCPs and 532.2 and 533.4 eV for BCPs-Ln3+-Phen, which are ascribed to O atoms of C=O and C-O, respectively. And the binding energy has a certain increase after coordination. These changes provide further evidence for N and O atoms coordination with Ln3+ ions.

Point 5:      p.7, line 203. I agree that the size of micelles is affected by the incorporated ion. The composition measurement is needed to explain about this.

Response 5:

Elemental analyses of the BCP-Ln3+-Phen with different molar ratios of Eu3+ and Tb3+ ions are also used to affirm the results. From Table 2, we can see the percentages of Eu3+ and Tb3+ atoms are essentially in agreement with the molar ratios of additions. The results confirm that the sizes of micelles are affected by the incorporated Ln3+ ions. 

Reviewer 2 Report

This article is an interesting research work. It is recommended for publication although it is advisable to make some changes or improvements, to facilitate its compression to the reader

Line 38.- In the introduction, he states that this experiment is interesting and inexpensive compared to other classic techniques such as doping with phosphorus. Should clarify or enhance these statements with references to studies that support it

Figure 1 It would be interesting to explain Figure 1 for an uninformed reader. A flow diagram of the process could clarify the figure better and reinforce the explanation of the text.

2.2 Characterization. Line 95 to 106. It should indicate the data of the Measuring Equipment in a Table attached to the Characteristics of the equipment used in the measurements.

Line 251 to 255. A table is recommended accompanying the figure. Could clarify with more precision the data of the results obtained.

Figure 8. The abacus color (c) of the figure points are not appreciated clearly. It is recommended to improve the quality of the graphics. A Table would improve the compression of the figure.

Conclusions.  An effort is requested to try to synthesize mehjor and summarize the results obtained. It is also advised to outline tips to promote lines of future research in line with the work carried out to deepen and improve the results obtained.

Bibliography. It is advisable to carry out a review of the bibliography in order to strengthen the study with references to more current works and located in other geographical places.

Author Response

This article is an interesting research work. It is recommended for publication although it is advisable to make some changes or improvements, to facilitate its compression to the reader

Point 1: Line 38. - In the introduction, he states that this experiment is interesting and inexpensive compared to other classic techniques such as doping with phosphorus. Should clarify or enhance these statements with references to studies that support it

Response 1: First one is blue-light as light-resource to excite yellow phosphors that is widely used due to the technique simplicity and cost lowness. The commendably realized white LED, YAG: Ce, is used as yellow phosphor. Unfortunately, it has many limitations, such as low luminous efficiency and color-rendering index [10]. Second one is UV-light excited phosphors to emit red, green and blue light, which can solve the above limitations and can improve the stability of light color. Whereas, for the later, there is extremely high requirement for the purity of red, green and blue emissions. Therefore, the scientific target is to find stable narrow emitting and high luminous efficient materials [11, 12].

10. Hu, Y.; Zhuang, W.; Ye, H.; Wang, D.; Zhang, S.; and Huang. X. A novel red phosphor for white light emitting diodes. J. Alloys Compd. 2005, 390, 226–229.

11. Ravindran, E.; Somanathan. N. Efficient white-light emission from a single polymer system with ‘‘spring-like’’ self-assemblies induced emission enhancement and intramolecular charge transfer characteristics. J .Mater. Chem. C, 2017, 5, 4763-4774.

12. Li, G.Q.; Wang, W.L.; Yang, W.J.; Wang, H.Y. Epitaxial growth of group III-nitride films by pulsed laser deposition and their use in the development of LED devices. Surf. Sci. Rep. 2015, 70, 380–423.

Point 2: Figure 1 would be interesting to explain Figure 1 for an uninformed reader. A flow diagram of the process could clarify the figure better and reinforce the explanation of the text.

Response 2: The BCPs are synthesized by RAFT polymerization. The rare earth ions are introduced into BCPs solution and coordinate with acrylic acid groups from different polymeric chains, leading to the formation of cross-linked PAA-Ln3+ domains. This reaction does not affect another segment, PS, to still remain its compatibility with the organic solvent. Consequently, BCP-Ln3+ micelles are formed with PAA-Ln3+ core and PS shell. Therefore, the stable micelles including Eu3+ and Tb3+ complexes with organic small conjugate ligands can provide tunable photoluminescence property to cover blue, green and red under a single 342 nm excitation. Tricolor lights can coordinate white light. So, BCPs-Ln3+-Phen emits bright white light.

Point 3: 2.2 Characterization. Line 95 to 106. It should indicate the data of the Measuring Equipment in a Table attached to the Characteristics of the equipment used in the measurements.

Response 3:

Table 1. Instrumentation is in the revised manuscript.

Point 4: Line 251 to 255. A table is recommended accompanying the figure. Could clarify with more precision the data of the results obtained.

Response 4:

Table 3. Decomposition temperature of BCPs and BCPs-Ln3+-Phen is in the revised manuscript.

Point 5: Figure 8. The abacus color (c) of the figure points are not appreciated clearly. It is recommended to improve the quality of the graphics. A Table would improve the compression of the figure.

Response 5:

Thank you for your advices, I have correccted the abacus color (c) of the figure points in the revised manuscript.

Table 4. CIE coordinates of BCPs-Ln3+-Phen under different excitation wavelength is in the revised manuscript.

Point 6: Conclusions.  An effort is requested to try to synthesize mehjor and summarize the results obtained. It is also advised to outline tips to promote lines of future research in line with the work carried out to deepen and improve the results obtained.

Response 6: This study provides a direction for further research of rare earth complexes in fully flexible white-light emitting diodes (FFWLEDs).

Point 7: Bibliography. It is advisable to carry out a review of the bibliography in order to strengthen the study with references to more current works and located in other geographical places.

Response 7:

16. Qian, W.; Zhang, A.; Wei, X. Structure and photoluminescence property of Eu, Tb, Zn-containing macromolecular complex for white light emission. Opt. Laser Technol. 2018, 107, 389-397.

39. Haldar, D.; Ghosh, A.; Bose, S. Defect induced photoluminescence in MoS2 quantum dots and effect of Eu3+/Tb3+ co-doping towards efficient white light emission, Opt. Mater. 2018, 79, 12-20.

40. Yahiaoui, Z.; Hassairi, M.A.; Dammak, M. Tunable luminescence and near white-light emission of YPO4:Eu3+, Tb3+, Tm3+, phosphors. J. Alloys Compd. 2018, 763, 56-61.

Reviewer 3 Report

Dear Authors,

The idea of energy transfer in such systems has been seen before. Here you say "The photoluminescent property of samples containing different molar ratio of Ln 3+ ions are
shown in Fig.7 (a) and 7 (b). We subsequently investigated the photoluminescent properties of Ln 3+
ion complexes and their counterparts by probing the 5 D 0 → 7 F 2 transition of Eu 3+ ions and 5 D 4 → 7 F 5
transition of Tb 3+ ions. Ln 3+ complexes have strong absorption in the range of 290 nm-360 nm ascribing
to the absorption of Phen ligand, which proves that energy can transfer from Phen ligand to Ln 3+ ions".

Can you prove it ? Do you say this based on other references ? Please put up some absorption and PL spectra to shwo the energy transfer. It is not possible that you can use the same material and excite directly the Ln ions? Is there an energy transfer among Ln ions at certain concentrations and above ?

You have discussed it somewhat in 3.1 but I do not really understand the text. There are many points to raise in 3.1. For example , Phen absorbs on its own, how do you make the Ln-Phen complex is strong or that is absorbs strongly?

Author Response

The idea of energy transfer in such systems has been seen before. Here you say "The photoluminescent property of samples containing different molar ratio of Ln 3+ ions are shown in Fig.7 (a) and 7 (b). We subsequently investigated the photoluminescent properties of Ln3+ ion complexes and their counterparts by probing the 5D0 → 7F2 transition of Eu3+ ions and 5D4 → 7F5 transition of Tb3+ ions. Ln3+ complexes have strong absorption in the range of 290 nm-360 nm ascribing to the absorption of Phen ligand, which proves that energy can transfer from Phen ligand to Ln3+ ions".

Point 1: Can you prove it? Do you say this based on other references? Please put up some absorption and PL spectra to show the energy transfer.

Response 1:  Excitation spectra is shown in (a) and the range of excitation wavelength is from 290 nm to 360nm. As well known, Ln3+ ions have extremely low molar absorption from 200 nm to 400 nm which is ascribed to the forbidden of electronic transitions between the 4f electronic states. Phen can be used as the “antenna” ligand due to its excellent molar absorptivity in the range. Because there is a suitable energy gap between trivalent state level of Phen and the Ln3+ ions emissive states, Ln3+ ions can effectively obtain energy from ligand under the action of appropriate antenna ligand.

Point 2: It is not possible that you can use the same material and excite directly the Ln ions?

Response 2: Trivalent state of Phen can match with the the trivalent ions' emitting levels of Eu3+ and Tb3+ ions. For this reason, we can just use Phen as the ligand.

Point 3: Is there an energy transfer among Ln ions at certain concentrations and above?

Response 3: When the spacing of Ln3+ ions is appropriate, there will be energy transfer between ions. We encapsulate Ln3+ ions with polymers to minimize energy transfer between them.

Point 4: You have discussed it somewhat in 3.1 but I do not really understand the text. There are many points to raise in 3.1. For example, Phen absorbs on its own, how do you make the Ln-Phen complex is strong or that is absorbs strongly?

Response 4: In the UV spectra, there is no contrast between the intensity of Phen and Ln3+ - Phen. We just use the red shift of absorption peaks to prove that Ln3+ ions coordinate with Phen. And I have revised points of 3.1.

The absorption band for BCPs ranges over 260 nm - 340 nm and the maximum absorption is at 268 nm, which is primarily attributed to the effect of the π-π* transition of benzene ring of PS-b-PAA [30]. Comparing the absorption band of Ln3+-BCPs with BCPs, the maximum absorption at 268nm becomes stronger. On the other hand, for free Phen, there is strong absorption band at 260 nm-340 nm including two absorption peaks at 294 nm and 324 nm respectively. They can be ascribed to π-π* transition and n-π* transition respectively [31]. The peak profiles of Ln3+-Phen and BCPs-Ln3+-Phen are almost similar to ligand Phen, excepting the red shift of their absorption peaks. The strong absorption peaks of Ln3+-Phen and BCPs-Ln3+-Phen are at 297 nm and 299 nm, respectively. Both of them have red shift compared with Phen which is attributed to the coordination of Ln3+ ions with N atoms of Phen. In addition, the strong absorption of BCPs-Ln3+-Phen, compared with extremely weak absorption of BCP-Ln3+, indicates that the UV-vis absorption mainly comes from Phen ligand.

Round  2

Reviewer 1 Report

The revised manuscript is acceptale for publication from Nanomaterials.